# Antitumor Efficacy of Focused Ultrasound-MFL Nanoparticles Combination Therapy in Mouse Breast Cancer Xenografts

**DOI:** 10.3390/ma13051099

**Published:** 2020-03-02

**Authors:** Daehyun Kim, Junhee Han, So Yeon Park, Heegon Kim, Ji-Ho Park, Hak Jong Lee

**Affiliations:** 1Department of Nano Science and Technology, Graduate School of Convergence Science and Technology, Seoul National University, Seoul 151-744, Korea; bluesolic@snu.ac.kr; 2Department of Radiology, Seoul National University Bundang Hospital, 82 Gumi-ro 173, Bundang-gu, Seongnam 13620, Korea; 3IMGT Co., Ltd., Seongnam 13605, Korea; soyeon.park@nanoimgt.com; 4Department of Bio and Brain Engineering and KAIST Institute for Health Science and Technology, Korea Advanced Institute of Science and Technology (KAIST), Daejeon 34141, Korea; mdjh1996@kaist.ac.kr (J.H.); heegon@kaist.ac.kr (H.K.)

**Keywords:** liposome, ultrasound, cancer

## Abstract

High doses of chemotherapy agents can cause adverse effects. To address this issue, drug-loaded vesicles with minimum drug loss, guided by an external element for precise delivery, are desired. Combinational therapy of both a focused ultrasound-induced drug delivery method and membrane fusogenic liposomes (MFLs) as drug delivery vehicles can satisfy such premises. In this study, we confirmed that the use of a small quantity of docetaxel-loaded membrane fusogenic liposomes (DTX-MFL) with focused ultrasound can induce better antitumor response in a xenograft mouse model compared to conventional docetaxel monotherapy and DTX-MFL only.

## 1. Introduction

Breast cancer still remains one of the common cancers in women, being ranked as the second-highest cancer in terms of occurrence. Despite its rate of high incidence, breast cancer patients’ 5-year survival rate is not the worst among cancers [1,2]. Although many efforts are spent in prevention and early diagnosis, patients often undergo mastectomy to remove one or both of their breasts and/or are treated with chemotherapeutic agents [3,4,5]. Classical chemotherapy relies on the systemic circulation and accumulation of cytotoxic chemical agents, which may not only reduce tumor growth but also damage normal cells as well [6,7]. Docetaxel, one of the most common drugs used against breast cancer, is normally administered in doses of 10–15 mg/kg in mice. Such a dosage is quite high when compared to other chemotherapeutics and therefore is known to cause detrimental side effects [8,9,10,11,12]. Therefore, researchers over the last few decades have been working on drug delivery techniques, including but not limited to production of vehicles that can carry and protect the therapeutic cargo and then subsequently release it at the target area to maximize drug efficacy while minimizing damage onto normal cells [13,14,15]. In this regard, the new method to synthesize liposomal nanoparticles, known as membrane fusogenic liposomes (MFLs), was invented [16,17,18]. Not only they are bio-compatible, they are subjected to the enhanced permeability and retention (EPR) effect and also can be tailored to fit desired characteristics. The main strength of MFLs is that they can be used to deliver both hydrophobic and hydrophilic compounds to cellular membranes and cytosol through fusion of the liposomal surface with the cellular membranes. Similar to conventional liposomes, MFLs are biocompatible and allow minimization of the drug dosage. Furthermore, the size of MFLs can be tailored to be around 100 nm, which allows these nanoparticles to migrate to the target site via the EPR effect in diseases such as solid tumors and atherosclerosis. In addition, rather than entering the cell through the endosome–lysosome pathway, MFLs may fuse with the cellular membrane, thereby efficiently delivering hydrophilic agents to target sites and yielding a therapeutic effect. Finally, functional modification can be applied to the lipids that constitute MFLs for the further targeting effect, including but not limited to conjugation of targeting moieties such as antibodies or peptides [17,18].

Along with advances in these drug delivery techniques, the use of ultrasound in drug delivery has been recently spotlighted [19]. Ultrasound can be applied as an external trigger to enhance drug delivery by the sonoporation effect [20,21]. Sonoporation is driven by the cavitation of microbubbles triggered by the ultrasound exposure. The repeated expansion and shrinkage of microbubbles, a phenomenon known as stable cavitation, creates microstreams that stress nearby cell membranes to induce disruption on vasculature. In addition, stronger ultrasound exposures eventually lead microbubbles to burst open. Known as inertial cavitation, this phenomenon creates pores of 100–300 nm on the cell membrane as the microbubble explodes and microjets and shock waves are generated [22]. This cavitation mechanism enhances drug delivery to the target area by improving uptake of drugs and drug vesicles [23,24]. In addition to normal ultrasound, focused ultrasound (FUS) has been proposed as another potential therapy option due to its unique ability to treat a specific region in the body without damaging nearby or intervening tissues and while also providing real-time monitoring of therapy [25].

In this study, we used a combination of FUS with MFLs containing a significantly lower amount of docetaxel and evaluated anti-cancer efficacy of this combination therapy in the MDA-MB-231 xenograft mouse model (Figure 1). MFLs loaded with 2 mg/kg of docetaxel were successfully prepared and characterized based on different features, including cell cytotoxicity and stability. Furthermore, their effectiveness as anti-cancer agents was tested in combination with FUS.

## 2. Materials and Method

### 2.1. Materials

The 1,2-dimyristoyl-sn-glycero-3-phosphocholine (DMPC), 1,2-dioleoyl-3-trimethylammonium-propane (DOTAP), and 1,2-distearoyl-sn-glycero-3-phosphoethanolamine-*N*-[methoxy(polyethylene glycol)-2000] were received from Avanti Polar Lipids (Avanti Polar Lipids Inc., Alabaster, AL, USA). Acetonitrile, chloroform, and methanol were received from Sigma Aldrich (Sigma Aldrich, St. Louis, MO, USA). The Zetasizer Nano ZS90 was received from Malvern Instruments (Malvern Instruments Ltd., Malvern, UK), and the 1260 Infinity II LC system was received from Agilent Technologies (Agilent Technologies Inc., Santa Clara, CA, USA). Docetaxel was received from MedChemExpress (MedChemExpress LLC, Monmouth Junction, NJ, USA). The VIFU 2000^®^ was received from Alpinion Medical systems (Alpinion Medical Systems Co., Ltd., Seoul, Korea). SonoVue^®^ was bought from Bracco (Bracco Imaging, Milan, Italy).

### 2.2. Preparation of Docetaxel-Loaded MFLs

Liposomal formulations were prepared from 1,2-dimyristoyl-sn-glycero-3-phosphocholine (DMPC, Avanti Polar Lipids), 1,2-dioleoyl-3-trimethylammonium-propane (DOTAP, Avanti Polar Lipids), 1,2-distearoyl-sn-glycero-3-phosphoethanolamine-*N*-[methoxy(polyethylene glycol)-2000] (DSPE-mPEG(2000), Avanti Polar Lipids), and docetaxel (Medchem express) using a film hydration/extrusion method according to the previous reports [17,18]. The molar ratios of DMPC, DOTAP, and DSPE-mPEG(2000) used for membrane fusogenic liposomes (MFLs) and non-fusogenic liposomes (NFLs) were 76.15:20:3.85 and 80:20:0, respectively. For liposomes loaded with fluorescent dye, 1,1′-dioctadecyl-3,3,3′,3′-tetramethylindocarbocyanine perchlorate (DiI, Invitrogen) was used. DiI-incorporated lipid film was prepared with 516.3 μg DMPC, 139.7 μg DOTAP, 108.0 μg DSPE-mPEG(2000) (for MFLs) or with 542.4 μg DMPC, 139.7 μg DOTAP (for NFLs), and 18.8 μg of DiI by dissolving them in organic solvents and then completely drying them overnight. The next day, the lipid film was hydrated using phosphate-buffered saline and then extruded through 100 nm membrane pores. For liposomes loaded with docetaxel, docetaxel was incorporated into the membrane of the liposomes using the hydration protocol. The docetaxel-incorporated lipid film was prepared with 516.3 μg DMPC, 139.7 μg DOTAP, 108.0 μg DSPE-mPEG(2000), and 40.39 μg of docetaxel by dissolving them in organic solvent and then completely drying them overnight. Lipid film was hydrated using phosphate-buffered saline and then extruded through 100 nm membrane pores. Docetaxel-loaded MFLs were stored at 4 °C until further use.

### 2.3. Characterization of MFLs

Hydrodynamic size, polydispersity, and zeta potential of prepared liposomes were measured using the dynamic light scattering (DLS) method (Zetasizer Nano ZS90; Malvern Instruments, Malvern, UK). In order to obtain the final concentration of docetaxel in liposomes, docetaxel concentrations were measured by high performance liquid chromatography (HPLC) (1260 Infinity ΙΙ LC system; Agilent Technologies, Santa Clara, CA, USA). A total of 100 μL of the docetaxel-loaded MFL solution was prepared and lyophilized (Modulspin 31; Hanil Science Medical, Korea). Then, it was suspended with 1 mL of methanol and sonicated for 10 min using a bath sonicator. After centrifugation at 15,000 rpm for 15 min, 500 μL of the supernatant was collected, and docetaxel remaining in the supernatant was analyzed by HPLC. The chromatographic conditions were as follows: Chromatographic separation was performed on a reversed phase C18 column. The compositions of the mobile phase were Acetonitrile/water (65:35, *v*/*v*) at a flow rate of 1 mL/min. Detection was taken at the wavelength of 230 nm. Encapsulation efficiency was calculated as the ratio of the amount of docetaxel into liposomes to the initial amount of drug. To test the stability of loaded cargos, the absorbance and fluorescence of DiI-loaded MFLs were measured using a UV–Vis spectrophotometer and spectrofluorometer (Molecular Devices, San Jose, CA, USA) (λ_ex_ = 530 nm and λ_em_ = 570 nm in fluorescence measurements). DiI-loaded MFL solution was filtered by centrifugal filter units (100 K MWCO, Millipore, MA, USA) to remove the leakage of DiI from the liposome at each time point. Fluorescence quantification was normalized by absorbance of lipids, and the remaining amounts of DiI were calculated.

### 2.4. Cell Culture

Human triple negative breast cancer cell line MDA-MB 231 cells were cultured in RPMI-1640 cell culture medium supplemented with 10% heat inactivated fetal bovine serum (FBS), 100 IU/mL penicillin, 100 mg/mL streptomycin, and 2 mM L-glutamine. Cultures were maintained in a humidified atmosphere with 5% CO_2_ at 37 °C. Cells were sub-cultured twice a week, with a seeding density of about 2 × 10^3^ cells/mL. Cell viability was determined by the trypan blue dye exclusion method.

### 2.5. In Vitro Fluorescence Cellular Imaging

To observe membrane fusogenicity of liposomes, 2 × 10^4^ cells of MDA-MB-231 cells were treated with medium containing 280μM of MFLs and NFLs loaded with fluorescent DiI for 15 min and further incubated for 30 min to 6 h at 37 °C. Cells were washed with PBS three times, stained with Hoechst 33342, and imaged using confocal microscopy (Nikon Instrument Inc., Tokyo, Japan).

### 2.6. Cell Viability-Assay

Cellular viability was examined using the MTT assay method. MDA-MB-231 cells were seeded to 96-well plates at a density of 2 × 10^3^ cells per well and left overnight in the incubator. The next day, these cells were treated with free docetaxel, MFLs, and docetaxel-loaded MFLs and were incubated for 48 h. Cells were then removed from the incubator, and their viability was evaluated using the MTT solution according to the manufacturer’s instruction.

### 2.7. In Vivo Study

The antitumor activity was evaluated using the MDA-MB-231 tumor-bearing BALB/C nude mouse model, which was established by a subcutaneous inoculation with the MDA-MB-231 cell suspension (1 × 10^6^ cells per mouse) into the right flank region of 4-week BALB/C nude female mice. After the tumor volume reached ~150 mm^3^, the mice were randomly sorted for treatment. Before the therapy, we examined MB + FUS to confirm that MB + FUS alone did not have anti-cancer effect. The experimental groups were defined as follows: (i) negative control, (ii) docetaxel only (2 mg/kg), (iii) docetaxel-loaded (DTX)-MFLs only (2 mg/kg), (iv) DTX-MFLs + MB + FUS. Sonovue MBs were injected with 1 mL per injection (1 × 10^8^ Sonovue MBs per mL). The tumor size was measured with a digital caliper, and volumes were calculated as width^2^ × length × 0.5. The cancer therapeutic analysis was determined based on tumor sizes from the date of first injection for each groups.

### 2.8. Focused Ultrasound (FUS) Treatment Parameters

A pre-clinical FUS system (VIFU 2000^®^, Alpinion Medical systems, Seoul, Korea) was used for ultrasound treatment. The therapeutic transducer used was a 1.1 MHz single-element spherical-focused transducer with a central circular opening of 40 mm in diameter. FUS exposure was performed after reaching the degassing level of ≤ 4 ppm. Prior to FUS treatment, intraperitoneal general anesthesia was administered using a mixture of 30 mg/kg zolazepam (Zoletil^®^, Virbac, Carros, France) and 10 mg/kg xylazine hydrochloride (Rompun 2%, Bayer Korea, Seoul, Korea). The tumor-bearing mice were set on a heating pad, and the target tumor was positioned at the center of the therapeutic transducer’s focal zone according to ultrasound guidance (E-CUBE 9^®^, Alpinion Medical Systems). The focal zone was 1.3 mm × 1.3 mm × 9.2 mm with a center frequency of 1.1 MHz at −6 dB. For precise targeting, the FUS system was equipped with 3D target position control (x-, y-, and z-axis). Pulsed FUS beams insonated the tumor and moved automatically at 2 mm space intervals to cover the entire tumor. The following acoustic parameters were used: frequency, 1.1 MHz; 20 Watts; pulse repetition frequency, 40 Hz; duty cycle, 5%; treatment duration, 5 s per spot; spot distance, 2 mm.

### 2.9. Statistical Analysis

All statistical analyses were performed using the ORIGIN 2020 software (OriginLab, OriginLab Corporation, Northampton, MA, USA). Data are expressed as mean ± standard deviation from at least 3 independent experiments. For in vitro experiments, statistical analysis of two groups was calculated with an unpaired two-tailed Student’s *t*-test. For in vivo animal test, one-way ANOVA was used for comparison between groups to determine the significance of the difference in tumor volume. Statistical significance was established as *p* < 0.05.

## 3. Results and Discussion

### 3.1. Characterization of MFLs

On the basis of previous reports [17,18], three types of lipids were used for synthesizing liposomes used in this study, and their physical characteristics are described in Table 1. Membrane fusogenic liposomes (MFLs) and docetaxel-loaded membrane fusogenic liposomes (DTX-MFLs) were first synthesized by the film hydration/extrusion method previously reported elsewhere, of which lipid compositions are listed in the Table 2. To measure the drug-loading efficiency, a loaded amount of docetaxel in 1 μmol of lipid was calculated from the value of the area under curve of the absorption peak of docetaxel by HPLC and the standard curve of the docetaxel concentration in methanol. As a result, it was calculated that 22.23 μg of docetaxel was loaded in 1 μmol of lipid, suggesting of moderate encapsulation efficiency (55.04%, Table 2).

Next, the physical characteristics of liposomal nanoparticles were measured. Hydrodynamic size of MFLs was measured to be 127.9 ± 2.2 nm according to the dynamic light scattering (DLS) measurements, while the size of DTX-MFLs was measured to be 125.2 ± 3.9 nm (Figure 2a,b). Surface charges of the nanoparticles were also measured by zeta potential measurement in DLS. Zeta potential of MFLs and DTX-MFLs was measured to be +24.5 ± 1.3 mV and +22.3 ± 2.2 mV, respectively (Figure 2c). To test their stability, size, and polydispersity index (PDI), DTX-MFLs were monitored until 48 h after synthesis. We found that there were no significant changes in the sizes and PDI of DTX-MFLs after 48 h. (Figure 2d,e). To test the stability of loaded cargos, DiI was used as a model hydrophobic cargo. Absorbance and fluorescence of DiI-loaded MFLs were monitored until 48 h after synthesis. After 48 h, about 90% of initially loaded DiI was finally left in MFLs, and we observed no significant cargo leakage (Figure 2f).

### 3.2. In Vitro Study

To evaluate the fusogenicity of MFLs, we first investigated whether MFLs could be transferred onto the membrane of target cells by fusion. Highly cationic non-fusogenic liposomes (NFLs) that are known to enter cells via endocytosis, the conventional pathway of nanoparticle uptake, were prepared for comparison (Table 2). MDA-MB-231 human breast cancer cells were treated with MFLs and NFLs loaded with hydrophobic fluorescent dye and DiI for 15 min and further incubated for 30 min to 6 h. As represented in Figure 3a, confocal microscopy revealed that DiI signals from the membrane of MFLs were efficiently transferred to the membrane of tumor cells. In contrast, cells treated with NFLs showed poor delivery of DiI onto the membranes, and observed DiI fluorescence was assumed to be transferred into subcellular compartments. After 1 h of incubation, cells treated with MFLs also showed delivery of DiI into the inner compartments. These results demonstrate that the liposomal membranes were successfully fused with cellular membranes, and MFLs were not attached to the outside of cellular membranes. Hydrophobic cargoes loaded in MFLs can enter the cytosol by incorporation into membranes of membrane vesicles (MVs), including exosomes and microvesicles. We speculate that the incorporation into the membrane of MVs can allow the enhanced penetration of hydrophobic drug cargoes, since MVs are known to play a key role in intercellular migration of exogenous hydrophobic cargoes through multiple cell layers [18].

Next, docetaxel, MFLs, and DTX-MFLs were evaluated for their potential anti-cancer effects. Docetaxel and DTX-MFLs showed significant growth inhibition/IC50 values against MDA-MB-231 human breast cancer cell lines for up to 24 h (Figure 3b). The DTX-MFLs group seemed to reach IC50 at a concentration of 40 nM and showed slightly more anti-cancer effect on the cells than docetaxel alone, while the empty MFLs vehicles did not have any effect on cell viability. However, anti-cancer efficacy did not exceed 50% because of the low dosage of docetaxel used. These results suggest that low doses of docetaxel do not elicit sufficient anti-cancer effects; therefore, it is important to improve the drug delivery technique to design the release of drugs specifically at the tumor area.

### 3.3. In Vivo Study

To evaluate the enhanced combination therapy effect of DTX-MFLs and FUS, tumor volumes were measured following the intravenous injection of PBS (negative control), free docetaxel, DTX-MFLs, or a combination of DTX-MFLs with MB and FUS. BALB/C nude mice xenografted with MDA-MB-231 cells were observed until the tumor volume reached around ~150 mm^3^ (N = 5 in each group). SonoVue^®^ (Bracco, Milano, Italy) commercially available clinically employed microbubbles, were injected intravenously prior to the injection of DTX-MFLs (Figure 4b). Sonovue MBs were evaluated prior to the in vivo therapy. The size of Sonovue MBs ranged from 1 to 10 µm, but over 90% of Sonovue MBs were in the range of 1 to 3 µm. The concentration of Sonovue MBs are around 1–5 × 10^8^ per mL. Between microbubble and DTX-MFLs injections, time interval was less than a minute. After the injection, FUS application was subsequently performed within a minute. Five injection days (0, 4, 8, 12, and 16 days) were planned. We also confirmed that the MB + FUS group did not show an anti-cancer effect, so MB and FUS were therefore categorized as positive controls (Figure 4a). It was found that the DTX-MFLs + MB + FUS injected group had significant inhibition of MDA-MB-231 tumor growth (*p* < 0.05) when compared with the control and free docetaxel groups. In addition, the free docetaxel group alone at the dosage used in our study did not have anti-tumor effects, as its tumor growths were comparable to the control groups. These data confirmed that the drug dose is a critical factor in treating cancer and that the amount of metabolized and excreted drugs from circulation is too high to affect the tumor growths in vivo. On the other hand, DTX-MFLs + MB + FUS treatment groups showed that the combination of focused ultrasound with MFLs allowed efficient delivery of docetaxel to the tumor without significant side effects. The preferential anti-cancer effect of MFLs can be explained by their strong membrane fusogenic ability, which allows delivery of the therapeutic payload at the target area while also minimizing drug loss on the way. Thus, despite using docetaxel at marginal amounts that do not affect tumor growth, DTX-MFLs were able to improve this issue so that even low doses could still exhibit a strong anti-cancer effect.

## 4. Conclusions

In summary, based on our data presented, MFLs as drug vehicles seemed to fuse well onto the cell membrane, allowing docetaxel to be delivered intracellularly to the tumor cells. Unlike free docetaxel, the DTX-MFLs’ formulation was able to evade clearance and move into the tumor region for higher exposure to the target, thereby allowing strong anti-cancer effects to occur despite using docetaxel in amounts lower than conventionally administered. In addition, FUS-induced microbubble cavitation seemed to cause sonoporation of the blood vessels, enhancing the EPR effect and allowing the membrane fusogenic liposomes to penetrate into the tumor area with higher efficiency. This strong anti-cancer mechanism therefore allows minimization of the side effects while greatly enhancing the drug efficacy at the same time.

## Figures and Tables

**Figure 1 materials-13-01099-f001:**
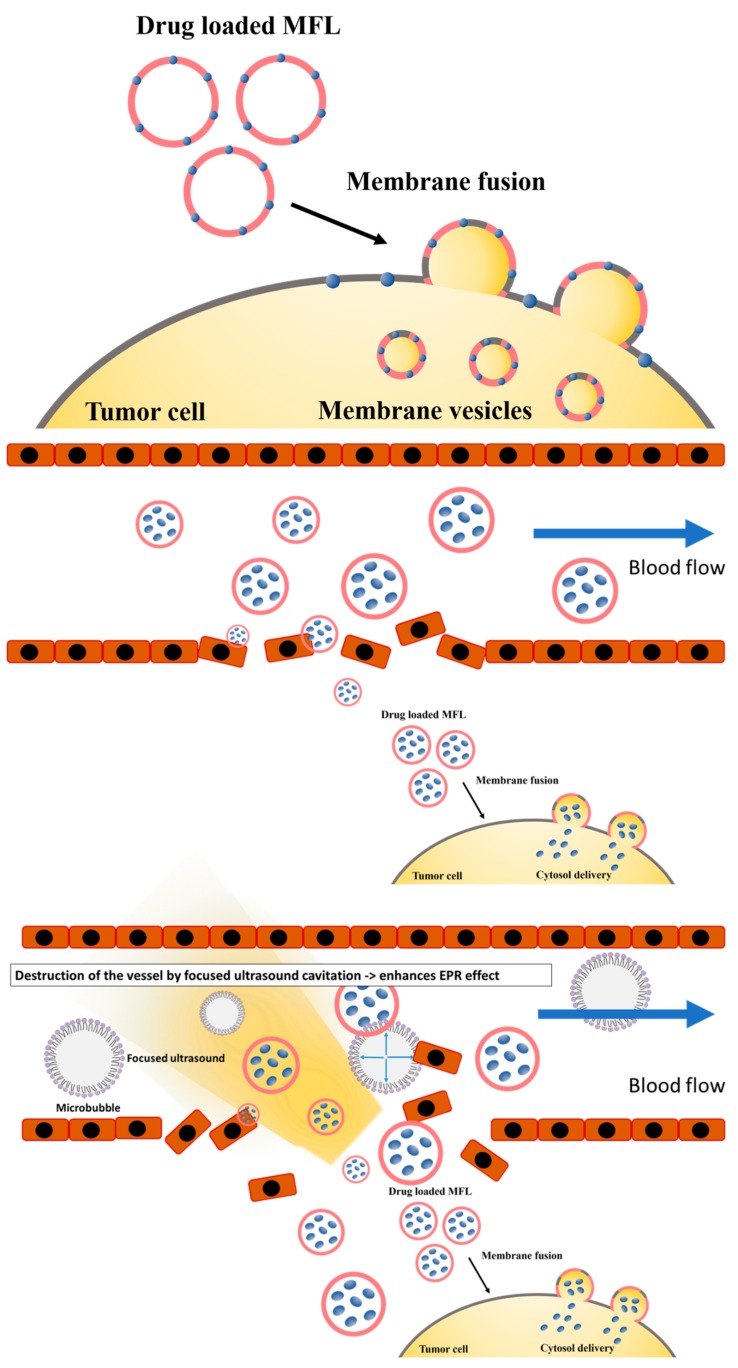
Graphical representation of membrane fusogenic liposomes (MFLs) and MFL-focused ultrasound.

**Figure 2 materials-13-01099-f002:**
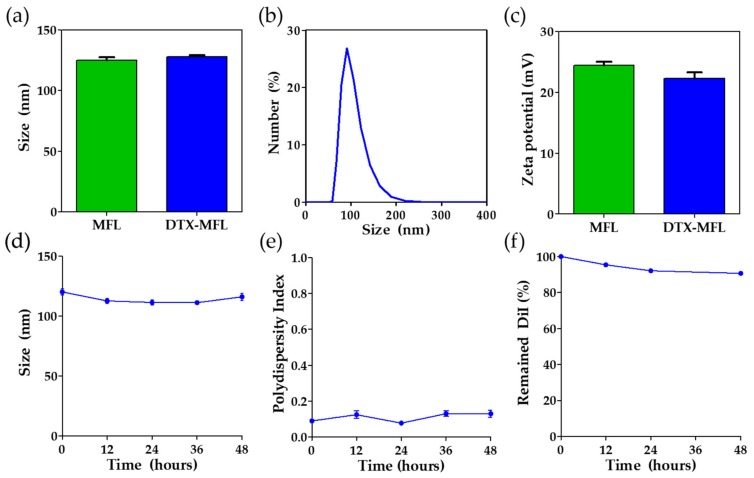
(**a**) Hydrodynamic sizes of MFLs and docetaxel-loaded (DTX)-MFLs as measured by dynamic light scattering (n = 4). (**b**) Distribution of hydrodynamic sizes of DTX-MFLs (n = 4). (**c**) Zeta potential of MFLs and DTX-MFLs (n = 6). (**d**) Particle size stability of DTX-MFLs up to 48 h. (**e**) Particle polydispersity index of DTX-MFLs up to 48 h. (**f**) Normalized fluorescence of remaining 1,1′-dioctadecyl-3,3,3′,3′-tetramethylindocarbocyanine perchlorate (DiI) in DiI-loaded MFLs up to 48 h. MFL and DTX-MFL denote membrane fusogenic liposomes and docetaxel-loaded membrane fusogenic liposomes. Data represent averages ± SD.

**Figure 3 materials-13-01099-f003:**
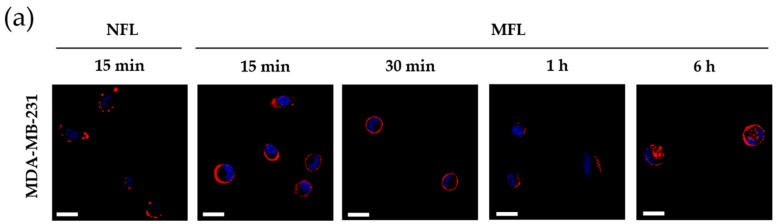
(**a**) Confocal fluorescent microscopic images of MDA-MB-231 cells treated with membrane fusogenic liposomes and non-fusogenic liposomes loaded with fluorescent dye DiI. (**b**) Cell viability of MFLs, free-DTX, and DTX-MFLs at 24 h. MFL and NFL denote membrane fusogenic liposomes and non-fusogenic liposomes, respectively. Nuclei were stained with Hoechst (blue). Scale bars represent 20 μm.

**Figure 4 materials-13-01099-f004:**
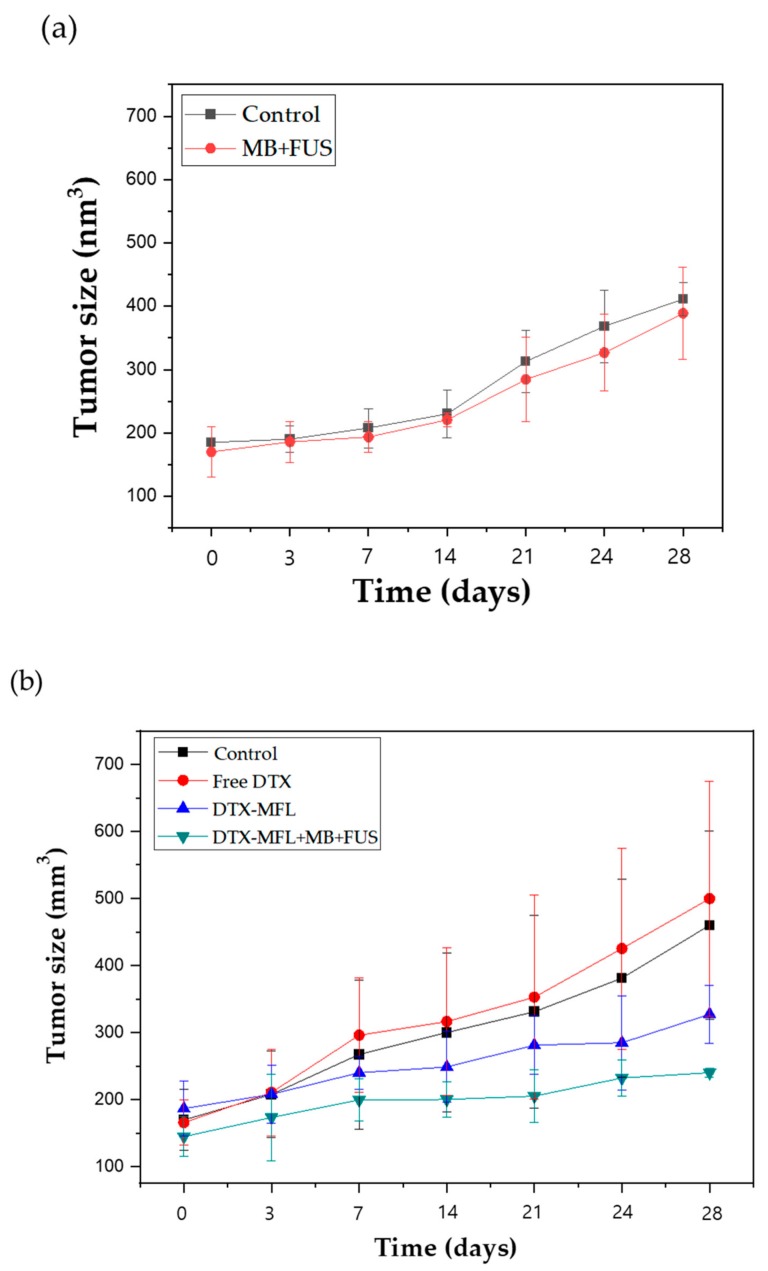
(**a**) In vivo efficacy of MB + FUS; (**b**) in vivo efficacy of docetaxel (free-DTX), docetaxel-loaded MFLs (DTX-MFLs), and docetaxel-loaded MFLs treated with microbubbles and focused ultrasound (DTX-MFLs + MB + FUS); (**c**) mouse body weight.

**Table 1 materials-13-01099-t001:** Information of lipids used for synthesizing liposomal formulations.

Full Name	Abbreviation	LipidChain	TransitionTemperature	Structure
1,2-dimyristoyl-sn-glycerol-3-phosphocholine	DMPC	14:0	24 °C	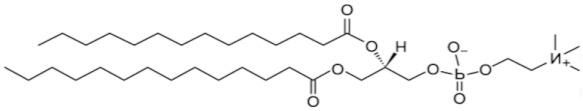
1,2-dioleoyl-3-trimethlammonium-propane	DOTAP	18:1	<5 °C	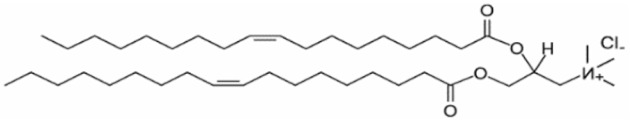
1,2-distearoyl-sn-glycerol-3-phosphoethanolamine-*N*-[methoxy(polyethylene glycol)-2000]	DSPE-mPEG(2000)	18:0	12.8 °C	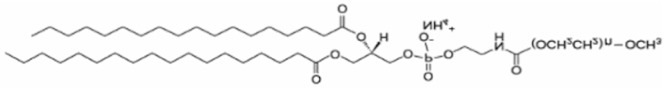

**Table 2 materials-13-01099-t002:** Lipid compositions and drug-loading efficiency of liposomes.

	Lipid Composition (Molar Ratio)				
Nanoparticle	DMPC	DOTAP	DSPE- mPEG(2000)	Docetaxel	Drug Loading Efficiency (%)
NFL	80	20	0	—	—
MFL	76.15	20	3.85	—	—
DTX-MFL	76.15	20	3.85	0.05	55.04

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
