# Peer review of "Antitumor Efficacy of Focused Ultrasound-MFL Nanoparticles Combination Therapy in Mouse Breast Cancer Xenografts"

_materials, 2020, doi:10.3390/ma13051099_

Round 1

Reviewer 1 Report

The work described the formulation of a liposomal drug delivery system for use in combination with focused ultrasound to deliver DTX to a breast cancer model. Due to substantial issues with both the written text and figure layouts as well as the experimental design I cannot recommend publishing the manuscript in its current form.

Substantial grammatical and spelling errors all throughout the text that negatively affects how the texts reads. These needs to be addressed.

Figures appeared with multiple layout and font types, sometimes making it hard to make out what is displayed on the axis (for example see Figure 3). Also, Figure 4 is missing a label on the x-axis making it impossible to interpret the results of this figure.

The authors are missing a control where 1) only FUS was applied and 2) FUS and free DTX was applied to ensure that the beneficial effect seen for the combination therapy is not solely related to the FUS treatment. Also, the authors state that the combination therapy displays significantly better results but do not provide any detail on the statistical test performed making it hard to evaluate this statement. In fact a visual inspection of the data does not a priori lend itself to the conclusion that there is any significance between the treatments (compare e.g. the blue and green in Figure 4) and the error on e.g. free DTX treatment seems extremely large (see at data point 7) almost spanning all other data sets. Taken together these concerns mean that form the current data set the conclusion made by the authors about a statistically significant therapeutic efficacy for the combination therapy are not supported. The statistical method used to evaluate this needs to be explained and/or the individual data sets could be provided.

The authors state that the data in figure 4 supports that the liposome fuses with the target cell membrane, however there is no evidence for this mechanism presented in this paper. Thus the mechanistic description described in figure 1 is purely speculation and needs to be tested in detail before the authors can claim that this is how the liposomes function. This also mean that the naming as “membrane fusogenic liposomes” (MFLs) is erroneous or at least not confirmed.

Author Response

First of all, 

We appreciate about the comments you gave us. It was good period of time that we had chance to look at our missing materials and data. Thank  you very much.

The errors that you mentioned had already been addressed in the paper and I am going to answer here as well (red letters below)

The work described the formulation of a liposomal drug delivery system for use in combination with focused ultrasound to deliver DTX to a breast cancer model. Due to substantial issues with both the written text and figure layouts as well as the experimental design I cannot recommend publishing the manuscript in its current form.

Substantial grammatical and spelling errors all throughout the text that negatively affects how the texts reads. These needs to be addressed.

-> The errors were addressed

Figures appeared with multiple layout and font types, sometimes making it hard to make out what is displayed on the axis (for example see Figure 3). Also, Figure 4 is missing a label on the x-axis making it impossible to interpret the results of this figure.

-> The errors were addressed

The authors are missing a control where 1) only FUS was applied and 2) FUS and free DTX was applied to ensure that the beneficial effect seen for the combination therapy is not solely related to the FUS treatment.

-> MB+FUS group was not placed in the data because such group was tested prior to the actual experiments to ensure that MB+FUS alone does not proide anti-cancer effect.
-> DTX+MB+FUS group was not al included in the study because the surfactant used for DTX destroys the lipid of MB, hence preventing further experiments with focused ultrasound

Also, the authors state that the combination therapy displays significantly better results but do not provide any detail on the statistical test performed making it hard to evaluate this statement.

-> The details on statistical tests were addressed in the materials and methods section

In fact a visual inspection of the data does not a priori lend itself to the conclusion that there is any significance between the treatments (compare e.g. the blue and green in Figure 4) and the error on e.g. free DTX treatment seems extremely large (see at data point 7) almost spanning all other data sets.

-> The difference between DTX-MFL and DTX-MFL+MB+FUS groups in Figure 4 were statistically significant. We do acknowledge the large error margin for DTX treatment-one of the mice treated with DTX did not have response and hence skwed our data, yet we kept it in our analysis for integrity. And we corrected it now.

Taken together these concerns mean that form the current data set the conclusion made by the authors about a statistically significant therapeutic efficacy for the combination therapy are not supported.

-> The details on statistical tests were addressed and we have obtained p value < 0.05 for the combinational therapy vs control, free DTX and DTX-MFL+MB+FUS

The statistical method used to evaluate this needs to be explained and/or the individual data sets could be provided.

-> The details on statistical tests were addressed in the materials and methods section

The authors state that the data in figure 4 supports that the liposome fuses with the target cell membrane, however there is no evidence for this mechanism presented in this paper. Thus the mechanistic description described in figure 1 is purely speculation and needs to be tested in detail before the authors can claim that this is how the liposomes function.

->We have included pictures taken with confocal microscopy that compares membrane fusogenic vs non-fusogenic liposomes. Non-fusogenic liposomes were designed to exclude PEG groups, hence giving the liposome a cationic character similar to common liposomes. These liposomes do not fuse onto the cell membrane, as observed by the confocal microscopy pictures.

This also mean that the naming as “membrane fusogenic liposomes” (MFLs) is erroneous or at least not confirmed.

-> The details on the naming of membrane fusogenic liposomes were addressed

Reviewer 2 Report

The subject of manuscripts is very relevant.
1) The inscriptions in the figures must be made larger and the same size.
In figure 3, the inscriptions are not visible.
2) Figure 2c is not needed at all, since it contains only 3 points. These data can be given in the text.
3) Explain which vessels are in mind when the penetration of liposomes between endothelial cells is described.

Author Response

First of all,

We appreciate about the comments you gave us. It was good period of time that we had chance to look at our missing materials and data. Thank you very much.

The errors that you mentioned had already been addressed in the paper and I am going to answer here as well (red letters below)

The subject of manuscripts is very relevant.
1) The inscriptions in the figures must be made larger and the same size.
In figure 3, the inscriptions are not visible.

-> The errors were addressed

2) Figure 2c is not needed at all, since it contains only 3 points. These data can be given in the text.

-> The errors were addressed

3) Explain which vessels are in mind when the penetration of liposomes between endothelial cells is described.

-> Blood vessels near the peripheral regions that are relatively permeable and vessels near interstitial regions that are often collapsed under high interstitial fluid pressure would be preferential for penetration of liposomes

And I am also attaching the file. 

Thank you 

Daehyun Kim

Reviewer 3 Report

The goal of this work was to maximize antitumor effects by using combination therapy with focused ultrasound-MFL nanoparticles in mouse breast cancer model. Authors claimed to produce membrane fusogenic liposome (MFLs) as drug delivery system for docetaxel.

FUS (focused ultrasound) is widely used clinically for drug delivery, fractured bone and cartilage, nerve stimulation, inflamed tendons and ligaments repairing. The non-thermal effects FUS mainly display in the mechanical stimulation induced by microbubbles, microjets, cavitation and acoustic steaming. Moreover, low-frequency FUS can reach deep into the body, which allows precise local treatment and avoids side effects of chemotherapeutic agents due to the use of lower doses.

Therefore, the topic of research is very interesting, unfortunately the way of implementation, presented experiments and their analysis need to be improved.

First of all, the Authors did not provide basic descriptions and structures of used compounds and particles. Full preparation of docetaxel-loaded MFLs is not presented, even not cited. Not all acoustic parameters are presented. What is pressure amplitude, pulse length, total exposure time? What is “20 Watts”? Authors meant 20 W/cm2? Authors used microbubbles (MB) but no MB parameters were presented, such as surface area, MB mean diameter, MB concentration. To achieve therapeutic effects, the amount injected may need to be increased to levels that should be tested for safety in blood circulation. What was the number of microbubbles per ml? Where other contrast-enhanced agents used for US? The Authors not presented detailed treatment protocols by using FUS. In the case of in vivo tests, Authors claimed that the experiment were performed when the tumors had grown to about 150mm3. How you calculated the size of the tumor? What was the period time between mouse inoculation and the initation of the experiment by using FUS? The “control” group is not correctly described. With consideration of safety was any other low acoustic intensity used? Or only 2.00 W/cm2? What was the protocol of histological examination? The discussion is presented briefly without careful analysis of the obtained results. Figure 3 – legend has to be improved The typos and errors in lines: 40, 106, 107, 117,118, 126.

 I recommend the manuscript for publication after a thoroughly improvement.

Author Response

First of all,

We appreciate about the comments you gave us. It was good period of time that we had chance to look at our missing materials and data. Thank you very much.

The errors that you mentioned had already been addressed in the paper and I am going to answer here as well (red letters below)

First of all, the Authors did not provide basic descriptions and structures of used compounds and particles. Full preparation of docetaxel-loaded MFLs is not presented, even not cited.

-> The errors were addressed

Not all acoustic parameters are presented. What is pressure amplitude, pulse length, total exposure time? What is “20 Watts”? Authors meant 20 W/cm2?

-> Watts indicate the total system input power from the transducer and W/cm2 is when the Watts is divided by the area which is called intensity. Pressure amplitude can be described as two categories; Peak Negative Pressure (PNP) is 3.64Mpa and the Peak Positive Pressure(PPP) is 4.54Mpa. We are not fully comprehending the question about pulse length but with our knowledge, when the transducer frequency is 1.1Mhz 1cycle per pulse length is 909ns and totla pulse length summation in one exposure time is 250ms.

Authors used microbubbles (MB) but no MB parameters were presented, such as surface area, MB mean diameter, MB concentration.

-> I used Sonovue which is commercially available but I guess that missing information could have confused the reviewer. 

->  The size of Sonovue MB is from 1 to 10 µm but over 90% of Sonovue MB is from 1~3µm. The concentration of Sonovue MB is around 1~5x108 per ml

To achieve therapeutic effects, the amount injected may need to be increased to levels that should be tested for safety in blood circulation. What was the number of microbubbles per ml? Where other contrast-enhanced agents used for US?

-> The main theme of this study was to lower the drug dose (compared to the conventional levels) yet still elicit anti-canceer effect so the question raised by reviewer " amount injected may need to be increased to levels that should be tested for safety in blood circulation." is unclear to the authors. Microbubbles were used according to the guidlines provided by the manufacturers.

-> The number of Sonovue microbubble are 1~5x108 per ml

The Authors not presented detailed treatment protocols by using FUS. In the case of in vivo tests,

-> MBs were injected first and MFLs next within a minute and applied the FUS right after injections were completed. Dtailed treatment protocols are written under "2.8 Focused ultrasound (FUS) treatment parameters."

Authors claimed that the experiment were performed when the tumors had grown to about 150mm3. How you calculated the size of the tumor?

-> The tumor volumes were measured with a digital caliper and were calculated as width2 x length x 0.5. The cancer therapeutic analysis was determined based on tumor sizes from the date of first injection on each groups.

What was the period time between mouse inoculation and the initation of the experiment by using FUS?

-> FUS treatment was performed immediately after injections were completed

The “control” group is not correctly described. With consideration of safety was any other low acoustic intensity used? Or only 2.00 W/cm2? What was the protocol of histological examination?

-> Based on our prior experiments on MB+FUS, no significant changes in the weight and behaviors of treatment mice were observed. Based on this data and analysis of literature, we took granted 2.00W as a "safe" point.

The discussion is presented briefly without careful analysis of the obtained results. Figure 3 – legend has to be improved The typos and errors in lines: 40, 106, 107, 117,118, 126.

-> The errors were addressed

 I recommend the manuscript for publication after a thoroughly improvement.

Round 2

Reviewer 1 Report

The work described the formulation of a liposomal drug delivery system for use in combination with focused ultrasound to deliver DTX to a breast cancer model. The revised manuscript represents a significant improvement, however I still have major issues that needs to be addressed before publication.

The overall level of the language has considerably improved in the revised version of the manuscript, but should still be upgraded before publication.

Also the scientific soundness has improved in the revised version, however I still have some major concerns that would need to be addressed before publication can be fully recommended.

There are some inconsistencies in how the authors describe their system and how it is depicted in figure 1. They say that the DTX is located in the membrane layer, but they depict it like it is in the encapsulated volume inside the liposome and delivered to the cytosol of the cell. Thus how the authors imagine the DTX reaching the cell interior is not obvious.

Liposome stability is a key feature in drug delivery, however the authors only provide minimal insights on this aspect. They state “To test their stability, size and polydispersity index (PDI) of DTX-MFLs were monitored until 48 hours after synthesis. We found that there were no significant changes in the sizes and PDI of DTX-MFLs after 48 hours.”.However these values can not be found anywhere in the manuscript and additionally, the most important parameter, the encapsulation efficiency does not seem to have been measured. Clear scientific evidence for these elements has be provided.

The added microscopy experiment depicted in figure 3a serves a valid control for MFL and NFL using different uptake mechanisms, however if DTX is delivered by fusion to the plasma membrane it would be beneficial to show contonous recrodings of the DiI to elucidate that it overtime will reach the inner comparmtents of the cell and don’t get stuck on the outside of the cells (e.g. if the MFL is only attached to the outside of the cell and not fused).

Since the stability of the encapsulation of DTX was not measured there is no control to validate that the effects seen in figure 4b is not just slow leaking liposomes. So to support the claim that it is the fusogenic nature the liposomes an not the fact that DTX is encapsulated, that are important it will be essential to see control with ‘non-fusogenic liposomes+MB+FUS’ and ‘free DTX+MB+FUS’.

The errorbars on the individual efficicay plots is also difficult to understand with a survival plot or a supplementary information figures of all the tumor growth curves for the individual mice. How can the errorbar on the control be extremely large, did some in this group show a decrease in tumor growth? And how can the error on the DTX-MFL+MB+FUS be so small, did the treatment halt tomour growth for all mice EXCANTLY in the same way?

The authors also don’t comment on the significance of the MB+FUS by comparing the DTX-MFL and DTX-MFL+MB+FUS treatments. They only focus on comparing DTX-MFL+MB+FUS with free DTX and control. Thus MB+FUS did have a beneficial effect is not know.

The authors still make unsupported claims like “Also, our data showed that FUS treatment induced microbubble cavitation and subsequent sonoporation effect, which led to increase in the penetration of vehicles into the tumor area and allowing more DTX-MFLSs to be fused with the target cell membrane for higher anti-cancer efficacy ”. They do not show increased carrier penetration into the tumor when using ultrasound, so this is speculation, what they show is decrease in tumor size. This effect could also be achieved is the MB+FUS treatment made the liposomes in the tumor unstable and thus made the rapidly release the DTX.

Author Response

Answers are written and some answers were corrected in the paper.

There are some inconsistencies in how the authors describe their system and how it is depicted in figure 1. They say that the DTX is located in the membrane layer, but they depict it like it is in the encapsulated volume inside the liposome and delivered to the cytosol of the cell. Thus how the authors imagine the DTX reaching the cell interior is not obvious.

->The errors were addressed

Liposome stability is a key feature in drug delivery, however the authors only provide minimal insights on this aspect. They state “To test their stability, size and polydispersity index (PDI) of DTX-MFLs were monitored until 48 hours after synthesis. We found that there were no significant changes in the sizes and PDI of DTX-MFLs after 48 hours.”.However these values can not be found anywhere in the manuscript and additionally, the most important parameter, the encapsulation efficiency does not seem to have been measured. Clear scientific evidence for these elements has be provided.

->The errors were addressed

The added microscopy experiment depicted in figure 3a serves a valid control for MFL and NFL using different uptake mechanisms, however if DTX is delivered by fusion to the plasma membrane it would be beneficial to show contonous recrodings of the DiI to elucidate that it overtime will reach the inner comparmtents of the cell and don’t get stuck on the outside of the cells (e.g. if the MFL is only attached to the outside of the cell and not fused).

-> The errors were addressed

Since the stability of the encapsulation of DTX was not measured there is no control to validate that the effects seen in figure 4b is not just slow leaking liposomes. So to support the claim that it is the fusogenic nature the liposomes an not the fact that DTX is encapsulated, that are important it will be essential to see control with ‘non-fusogenic liposomes+MB+FUS’ and ‘free DTX+MB+FUS’.

->The errors were addressed

The errorbars on the individual efficicay plots is also difficult to understand with a survival plot or a supplementary information figures of all the tumor growth curves for the individual mice. How can the errorbar on the control be extremely large, did some in this group show a decrease in tumor growth? And how can the error on the DTX-MFL+MB+FUS be so small, did the treatment halt tomour growth for all mice EXCANTLY in the same way?

-> I did not consider that fact seriously or susceptive because the it was control group. I could have taken out few outliers from the data in control group which I might have needed. If I can explain about the error bar issue, let me put this in the logic that might be explained: I let the tumor cell growth freely (control) -> some cancer cells growth fast but some did not(especially MDA-MB-231 in my experience). However, If I treated with strong chemodrug like DTX(treatment group), every cancer cell can be killed, this (This can explain the error bar issue). And for the last question, I guarantee that the treatment halt tumor growth for all mice EXACLTY in the same way.

The authors also don’t comment on the significance of the MB+FUS by comparing the DTX-MFL and DTX-MFL+MB+FUS treatments. They only focus on comparing DTX-MFL+MB+FUS with free DTX and control. Thus MB+FUS did have a beneficial effect is not know.

-> I would ask to be excused if I did not comment on the significance of the MB+FUS compared to other groups. The reason I did not mention is that the MB+FUS group is very similar to the Control group. I tried to highlight that there is no difference between MB+FUS and Control groups and I think it is unnecessary for additional comparison. 

The authors still make unsupported claims like “Also, our data showed that FUS treatment induced microbubble cavitation and subsequent sonoporation effect, which led to increase in the penetration of vehicles into the tumor area and allowing more DTX-MFLSs to be fused with the target cell membrane for higher anti-cancer efficacy ”. They do not show increased carrier penetration into the tumor when using ultrasound, so this is speculation, what they show is decrease in tumor size. This effect could also be achieved is the MB+FUS treatment made the liposomes in the tumor unstable and thus made the rapidly release the DTX.

-> I think decrease in tumor size is the most effective reason to explain this phenomenon. I strongly do not think that ultrasound with that pressure can burst the liposome but the microbubble unless the ultrasound is too high to burn the tumor area which I did not observe while treatment. Therefore I used membrane fusogenic liposome which was already been tested in in vitro study and the cavitation effect can surely induce the MFLs to the tumor area and the accumulation of MFLs was increased and MFLs was going to be fused to the cell that cause the anti-cancer effect.

I really appreciate that reviewer woke me to think that I need to study further relate to this field for the next research article. Study relationship and details more into the mechanisms of drug delivery can be another solid article. 

There are some points that are not obvious in this articles, however, I will sure study more into the facts and wanting to publish another one about this issues.

Thank you for pointing me out. 

Reviewer 3 Report

After reading the authors' answers, I recommend the article for publication